# Airway Management in Pediatric Patients: Cuff-Solved Problem?

**DOI:** 10.3390/children9101490

**Published:** 2022-09-28

**Authors:** Eva Klabusayová, Jozef Klučka, Milan Kratochvíl, Tereza Musilová, Václav Vafek, Tamara Skříšovská, Jana Djakow, Martina Kosinová, Pavla Havránková, Petr Štourač

**Affiliations:** 1Department of Paediatric Anaesthesiology and Intensive Care Medicine, University Hospital Brno and Faculty of Medicine, Masaryk University, Kamenice 5, 625 00 Brno, Czech Republic; 2Department of Simulation Medicine, Faculty of Medicine, Masaryk University, Kamenice 5, 625 00 Brno, Czech Republic; 3Paediatric Intensive Care Unit, NH Hospital Inc., 268 31 Hořovice, Czech Republic; 4Department of Anaesthesiology and Intensive Care Medicine, The Donaustadt Clinic, Lango Bardenstraße 122, 1220 Vienna, Austria

**Keywords:** cuff, airway management, pediatric patient, uncuffed tube, cuffed tube

## Abstract

Traditionally, uncuffed tubes were used in pediatric patients under 8 years in pursuit of reducing the risk of postextubation stridor. Although computed tomography and magnetic resonance imaging studies confirmed that the subglottic area remains the narrowest part of pediatric airway, the use of uncuffed tubes failed to reduce the risk of subglottic swelling. Properly used cuffed tubes (correct size and correct cuff management) are currently recommended as the first option in emergency, anesthesiology and intensive care in all pediatric patients. Clinical practice particularly in the intensive care area remains variable. This review aims to analyze the current recommendation for airway management in children in emergency, anesthesiology and intensive care settings.

## 1. Introduction

Based on historical anatomical studies of the pediatric larynx [1,2], uncuffed tubes were preferred in pediatric patients under 8 years [3,4]. The main aim was to obtain a proper seal of the airways in the narrowest part—the subglottic area—without direct pressure on the airway mucosa to minimize possible mucosal damage, which could lead to consequent edema formation and postextubation stridor or even laryngeal damage. Therefore, in an ideal situation, uncuffed tubes should be a safer alternative to cuffed tubes, predominantly used in older children and adults. The superior safety has never been proven by high-quality evidence-based medicine (EBM); however, for a long time, it remained the unquestionable rule of pediatric airway management. In the recent two decades, several articles revealing the lower incidence of postextubation stridor with cuffed tubes have been published [5,6,7]. Extensive anatomical research (real-time bronchoscopy, computed tomography and magnetic resonance imaging) revealed that the originally thought circular-shaped subglottic area is in reality elliptical, with a wide anteroposterior diameter and narrow transverse diameter [8,9,10,11]. These findings could explain the risk of postextubation stridor even with the properly placed uncuffed tube, which could severely damage the mucosa in the transverse edges of the subglottic region. Specifically designed cuffed tubes for pediatric patients (e.g., Microcuff^®^) with the redesigned high-volume low-pressure cuff (smaller and cylinder-shaped) and with the ultrathin cuff membrane from polyurethane, located more distally on the tube without Murphy’s eye, have been developed for pediatric airway management in early 2000 and were intended for pediatric patients ≥3 kg of body weight [12]. Cuffed tubes, due to several advantages over uncuffed tubes (lower reintubation rates, better seal, lower risk of aspiration, capnography tracing, optimal ventilation management, etc.) [5,6,13,14,15,16,17,18], gained significant popularity in pediatric anesthesiology in recent decades and their use has an uprising tendency also in pediatric intensive care settings [19,20,21,22,23]. Cuffed tubes should now be preferred for all children ≥3 kg of body weight in the emergency setting [24] and anesthesia airway management [25]. According to Kneyber et al. [26], cuffed tubes could be safely used also in the pediatric intensive care setting with adherence to proper cuff management (monitoring the intra-cuff pressure and limiting the cuff pressure to ≤20 cm H_2_O) [26]. However, the paradigm shift has been adopted with significant personal and regional variability. Currently, many pediatric anesthesiology and intensive care centers are still preferably using uncuffed tubes. This review article aims to discuss the available evidence, considering the benefits and possible risks of cuffed tubes in pediatric emergency, anesthesiology and intensive care settings.

## 2. Airway Anatomy

Several differences have been described between the airway anatomy in adults and pediatric patients, with the most considerable ones in neonates. The larynx in children is located more cranial (*rima glottidis* at the level of C3), and as the child grows, the position shifts to the C4 level and, finally, to the C4–C5 level [3]. The epiglottis is longer and U-shaped. The narrowest part of the pediatric airway seems to be the subglottic area [11], However, Dalal et al. described the *rima glottidis* (as in adults) as the narrowest part [27]. The difference probably lies in the distensibility of these areas. The subglottic area at the level of the cricoid ring is stiff and nondistendable, while the glottis area is distendable manually and even pharmacologically (muscle relaxation). After a detailed computed tomography evaluation of the pediatric airway, significant differences between the infraglottic area and the airway area at the level of cricoid were found [11]. These levels were traditionally defined as one, but according to Mizuguchi et al., the pediatric airways at the level of cricoid are indeed circular, as described previously [1,2]; however, the narrowest part of the airway lies at the infraglottic level (slightly above the cricoid cartilage) [11]. The infraglottic area was elliptical in all analyzed patients (n = 86), with a short transverse diameter. The most important finding is that the transverse diameter at the infraglottic level was always smaller than the diameter at the cricoid level [11].

## 3. Uncuffed vs. Cuffed Tubes

The recommendation for uncuffed tubes is to choose the size that will pass through the subglottic area and lead to the adequate seal of the system without damaging the airway mucosa. The aim to reach the seal up to 20–30 cm H_2_O of peak inspiratory pressure [8], and the leak over 20–30 cm H_2_O was considered as safety aspect (explained by a certain superstition that the leak will guarantee no/minimal tube-to-mucosa contact/damage). This, however, could not be carried out effectively/safely due to the airway anatomy described above (elliptical shape of subglottic area). Nevertheless, uncuffed tubes still have their place in neonatology—in low birth weight and extreme low birth weight neonates—due to the absence of 2.5 cuffed tubes on the market.

Cuffed tubes lead to improved seal compared to their uncuffed equivalents, lower exchange rate, accurate capnography and spirometry tracing, decreased incidence of pulmonary aspiration, minor postoperative complications and lower incidence of sore throat [8,27]. Pediatric cuffed tubes were originally manufactured by downsizing the adult tubes. This practice unfortunately did not respect the difference in pediatric airway anatomy. The large cuffs made of polyvinylchloride (PVC) were replaced by an ultrathin membrane from polyurethane, and the cuff has been redesigned (smaller, cylindrical) and relocated more distally on the tracheal tube. Due to the small distance from the subglottic area to the tracheal bifurcation in neonates and infants, the Murphy’s eye was dismantled. Additional attention has been dedicated to the cuff pressure monitoring/management, aiming to minimize the intracuff pressure to minimal possible pressure that sealed the system effectively and, at the same time, to always keep the pressure lower than 20 cm H_2_O. According to the published data, the intracuff pressure for an adequate seal could be reached with a pressure at 10 ± 4.3 cm H_2_O [6,28,29]. There have been several methods describing how to measure intracuff pressure (e.g., pilot balloon finger palpation, minimal occlusive volume and minimal audible leak) [30]. Especially for ICU settings in intubated patients, where manual monitoring of cuff pressure needs to be performed repeatedly in order to achieve a range of safe pressures, a continuous cuff pressure monitoring system (e.g., IntelliCuff^®^) can be used (Figure 1). The device continuously keeps the pressure in the cuff in a set range. The aim of strict cuff pressure measurement is not only to prevent mechanical tracheal injuries caused by an overdistended cuff but also to lower the risk of ventilator-associated pneumonia [31].

## 4. Tube Size

The proper size of the tracheal tube for an adult patient is based predominantly on gender. In average female, cuffed tubes sized between 6.5 and 8, and in average male, size 8.5–9.5 are being used. Due to airway width, there is only a little fear of postextubation stridor in adults and only cuffed tubes are worldwide used for patients over 18 years. The narrowest part of the airway in adults is the *rima glottidis*, so the size of the tube should be adjusted according to the laryngoscopy view during the intubation. In pediatric patients, cuffed, uncuffed tubes and specialized cuffed tubes (e.g., Microcuff^®^) are currently being used for tracheal intubation. The proper size of the tube could be based on the patients age by Cole age-based formula for uncuffed tubes (internal diameter − tube size = age in years/4 + 4) [11,32] or Motoyama formula [33] for cuffed tubes (internal diameter − tube size = age in years/4 + 3.5), however only with 50–75% accuracy [8,32,34,35]. The risk of oversizing the tube size based on these formulas in patients under 10 years is significantly higher for uncuffed tubes in comparison to cuffed tubes (60.0% vs. 23.8%, *p* < 0.05) [11]. Nevertheless, these age-based formulas do not take the patient’s weight and height into consideration. In combination with the narrowest subglottic area that cannot be visualized during the direct laryngoscopy, the proper tube size prediction in pediatric patients is somewhat limited. Tube selection by the terminal part of the index finger phalanx or the little finger phalanx could be used in an emergency setting but seems to be highly unreliable [31].

When using the Microcuff^®^ tubes, the proper sizing is based on the patient’s age and derived from manufacturer recommendations. The smallest Microcuff^®^ tube currently available on the market is 3.0, recommended for pediatric patients ≥3 kg of body weight and aged between 0–8 months; however, in the close future, 2.5 tubes are going to be introduced to the market. The recommended tube size for pediatric patients based on Cole’s and Motoyama’s formula together with the Microcuff size selection is listed in Table 1.

The possible method for tube size prediction seems to be the ultrasound bedside airway dimension measurement [36]. Several studies have already confirmed the usefulness of ultrasound for estimating adequate endotracheal cannula size compared to age-based formulas. Schramm et al. demonstrated a strong correlation between the outer diameter of the uncuffed endotracheal tube with the minimal transverse diameter of the subglottic airway measured by ultrasound and showed a reduction in the number of reintubations [34,36]. Other studies demonstrated that ultrasound tube size prediction is more accurate than age-based formulas, as the latter significantly overestimated the tube size [34,37]. These findings are similar when comparing the ultrasound estimated tube size with both Cole’s and Motoyama’s formula [37,38]. The subglottic diameter is best measured in anesthetised and paralyzed patients, as the phase of respiration may affect the diameter of the area. This is also the reason why, in comparison to adults, the ultrasound is not yet routinely used for the prediction of difficult airways, as it needs to be performed before the anesthesia induction necessitating a cooperative patient [39]. Ultrasound assessment was also successfully used to confirm tracheal intubation as well as the correct depth of endotracheal cannula insertion [40]; however, there are no data that evaluate its use for assessing the front of neck access (FONA).

## 5. Airway Management in Pediatric Anesthesiology

Cuffed tubes have become preferred in all pediatric patients, including patients under 8 years in the recent decade [41]. According to the survey from 2016, 85% of pediatric anesthesiologists preferred cuffed tubes for children over 2 years and 60% for full-term neonates [42]. After repeatedly proven to be superior with a lower exchange rate [7], superior ETCO_2_ tracing, spirometry, lower leak together with a never proven risk of higher sore throat incidence [43] and/or postextubation stridor incidence [44], the only remaining reasons uncuffed tubes are used are tradition and lower costs. According to the EBM data, it currently seems reasonable to avoid uncuffed tubes in pediatric anesthesia completely, with the exception being where the outer or inner diameter of the tube play crucial roles (e.g., the diameter needed for bronchoscopy passage or in neonates with low and extremely low birth weight). Another aspect that strengthens the cuffed tube’s position in pediatric airway management is the SARS-CoV-2 pandemic, where maximum effort has been made to minimize the aerosol spread during airway management. Preoxygenation together with rapid sequence induction and tracheal intubation with cuffed tubes has been recommended for all adult and pediatric intubations [25,45].

Effective sealing of the airway provided by the cuff might be of great benefit for example in procedures where there is raised intraabdominal pressure and altered diaphragm position (laparoscopy and thoracoscopy). In these cases, the more effective seal and no leak can prevent difficulties in ventilation with raised peak pressures, and moreover, it provides better protection against a leakage of stomach contents into the airways [3]. In case of procedures where there is a risk of airway fire (procedures on the oropharynx, e.g., adenotonsillectomy), the proper seal of the cuff can prevent contamination of the oropharynx with the inhaled gas mixture with often higher oxygen concentrations [3].

Low birth weight neonates however remain an indication of uncuffed tubes, due to the absence of a cuffed tube smaller than 3.0 on the market. In the small retrospective trial analyzing the data from 23 neonatal intensive care unit (NICU) patients, Thomas et al. found that the limit for effective use of a 3.0 cuffed tube could reach <3000 g [12]. The limit of >2700 g body weight (to reach >50% efficacy) was established in 2021 by Zander et al. [46] after analyzing the data from 269 neonates requiring intubation by a pediatric anesthesiologist with 3.0 cuffed tube Microcuff^®^. For neonates with a weight below 2700 g, the outer tracheal tube diameter is currently the limitation for the cuff; however, as 2.5 size cuffed tube development has been already announced, this could soon lower the weight limit even more. These limits are based on analyzing the most prevalent neonatal/pediatric cuff tube—Microcuff^®^—but also different limits could be clinically relevant when using cuffed tubes from different manufacturers due to wide outer cuff diameters differences (lower, with polyurethane cuff 91–118% compared to polyvinylchloride cuff 91–146%) [47]. Therefore, comparing the pros and cons of cuffed tubes in anesthesiology, the uncuffed tubes should be abandoned in all cases, where cuffed tubes can be used.

Contrary to PICU/NICU settings, in anesthesiology settings, a whole spectrum of supraglottic airway devices has been developed over the past four decades, with the laryngeal mask being the most prevalent. As supraglottic devices have been successfully used in a whole spectrum of adult and pediatric anesthesiology sections (including surgeries in prone position, laparoscopic surgeries, etc.) [48,49,50,51] with comparable efficacy and lower incidence of associated complications in comparison to tracheal tubes [52,53], the supraglottic airways should always be considered as the device of choice in all situations possible (fasted patient, low risk of aspiration/regurgitation, and possible access to head and airway of the patient). Currently, multiple types of laryngeal masks are available on the market, but the majority are equipped with an inflatable cuff to obtain an adequate perilaryngeal seal (cuff located in the hypopharynx) after inflation [54]. The proper cuff management is vital also in case of laryngeal mask cases due to the risk of oropharyngeal and hypopharyngeal mucosal cuff-related pressure damage. According to the manufacturer’s recommendations, the optimal laryngeal mask intracuff pressure should be between 40 and 60 cm H_2_O, although this level of pressure could reduce the capillary flow in the surrounding tissue and eventually lead to sore throat, cough and airway-related morbidity after the emergence form anesthesia. It seems reasonable to lower the intracuff pressure below 40 cm H_2_O, and sufficient seal could be achieved also with pressure between 20 and 40 cm H_2_O.

## 6. Airway Management in Neonatal Intensive Care (NICU) and Pediatric Intensive Care (PICU)

In pediatric intensive care, the conversion from uncuffed to cuffed tubes is still an ongoing process. In 2013, data showed that, in some PICUs, cuffed tubes have already been the preferred choice for intubation [22], but with significant regional variability and slower transition in neonatal intensive care units. When considering the regional variability of cuffed tubes in intensive care, according to the Scandinavian survey from 2015 [55], 50% of PICUs were using cuffed tubes, compared to 100% of PICUs and 33% of NICUs in Australia and New Zealand in the same year [20,23]. It seems that cuffed tubes are being used more frequently in ICUs which are run by anesthesiologists compared to pediatricians and/or neonatologists [23,55]. This could be explained by the experience of transition from anesthesia care, where cuffed tubes are being used for a longer time. In intensive care settings, infectious complications in critically ill patients are frequent and incidence rise linearly with the length of stay. One of the most feared and prevalent infection in ICU/PICU settings is ventilator-associated pneumonia (VAP). It is defined as the development of pneumonia after 48 h from intubation in a mechanically ventilated patient [56,57]. VAP development is strongly associated with prolonged length of mechanical ventilation, ICU stay, in-hospital stay, morbidity, mortality and overall costs [58,59,60]. Ventilator-associated pneumonia is a common nosocomial infection associated with ventilated patients, and the mortality of patients suffering from VAP is high. Bacterial colonization and repeated microaspiration have been described as the main pathophysiologic mechanism of VAP development [61]. Based on these findings, the proper seal of the airway based on an optimally managed cuffed tracheal tube could significantly reduce the incidence of ventilator-associated pneumonia compared to uncuffed tubes, nevertheless, no high-quality EBM data have been published considering this topic in pediatric patients. The problem of VAP could, however, only be partially solved with the cuff when analyzing the high incidence (up to 13–51 per 1000 ventilatory days) of VAP in adult ICUs (with only cuffed tracheal tubes) [62]. Of several risk factors for VAP such as a nasogastric tube, sinusitis, neutral patient position, upper respiratory tract bacterial colonization, biofilm, open suctioning, gastric acid suppression treatment (proton pump inhibitors and/or H2 blockers) and cuff management, precisely the incidence and time spent below 20 cm H_2_O of intracuff pressure has been associated with higher VAP incidence [62,63]. These findings, though outdated and based on adult patient´s data, raise the question of the optimal intracuff pressure in children to balance the risk of microaspiration (with lower pressure) and the risk of mucosal damage (in pressure above 20 cm H_2_O). Despite proper seal, microaspiration could still occur. The design of the tube (high-volume low-pressure) and the cuff material seem to have a protective effect. Furthermore, polyurethane cuff (e.g., Microcuff ^®^) has been associated with lower pericuff leak of blue dye [57,64,65,66]. Based on the pathophysiology of VAP, cuffed tubes with the possibility of subglottic secretion suctioning/clearance have been introduced into clinical practice in both adults and children—Figure 2. Despite the cuffed tubes boom in the recent decade, only limited data are available considering their long-term use [47,67], although no data reported higher complication rates in ICU/NICU setting.

## 7. Cuffed Tracheostomy Tubes

Although minimally discussed or published, the same uncuffed vs. cuffed controversy also arose in patients with tracheostomy tubes. Tracheostomy in PICU or NICU setting is indicated in patients with repeatedly failed weaning, dependent on mechanical ventilation. The aim is to obtain a secured entrance to the airway with an infraglottic wide tracheostomy tube, which allows for invasive suctioning of the airways, limiting the undesired protective airway reflexes caused by the tracheal tube lying in between vocal cords and significantly reducing the airway resistance that could ameliorate the weaning process. The tracheostomy could be introduced percutaneously (percutaneous- dilatation tracheostomy—PDT) or traditionally by surgical intervention. Based on adult data, PDT is associated with a lower incidence of complications [62], less time and lower costs [68]. However, tracheostomy in adult intensive care is much more prevalent compared to PICU/NICU settings, and the PDT is very rarely used in children due to anatomical reasons. Historically, uncuffed tracheostomy tubes were preferred in children under 8 years old due to fear of possible pressure damage to the mucosa in the infraglottic region. Traditional teaching aiming to protect the vulnerable narrowest part of the infraglotic area (subglottic area) has no use in this case because tracheostomy usually lies below this area. The EBM evidence of what is good practice when it comes to tracheostomy in children is even scarcer than the evidence for tracheal tubes. The advantages of cuffed tracheostomy tubes are similar to those mentioned for the tracheal tubes, while the most common and feared complications are not relevant here as the subglottic area is not involved when it comes to the tracheostomy tubes. Especially for children on long-term mechanical ventilation, there is generally no proven reason to avoid cuffed tracheostomy tubes in any age group. However, uncuffed tracheostomy tubes still might have their place in some patients during the weaning from the mechanical ventilation or in some spontaneously breathing patients before tracheostomy extubation.

## 8. Cuff-Related Complications

One of the most feared yet never EBM-proven complications of the cuffed tube is considered the subglottic laryngeal edema, with a postextubation stridor that could ultimately lead to laryngeal stenosis formation. Postextubation stridor (PES) is strongly associated with airway-related morbidity and even mortality in pediatric anesthesiology and the pediatric intensive care unit (PICU) [69]. The reported incidence of PES in PICU settings is between 6 and 30% [70,71,72]. The risk factors for PES development have been identified as the length of intubation and size of the tube, but not the cuff itself [6,14,69,71,72]. The safety factor seems to be the proper intracuff management and limiting the pressure to ≤20 cm H_2_O [26], ideally to the minimal pressure needed for a proper seal of the system (<15 cm H_2_O for Microcuff^®^ tubes seems to be sufficient) [73]. Without quantitative intracuff pressure monitoring, pressure over 90 cm H_2_O had been frequently detected [74,75]. Due to the high incidence of PES, routine prophylactic corticosteroid pretreatment had been adopted in many PICUs and anesthesiology departments prior to extubation with a proven positive effect on PES reduction [76]. The ultrasound evaluation of air column width before and after cuff deflation may be used in the prediction of PES, as it showed high accuracy [77]. In the prediction of PES, this suggests possible superiority in comparison with the traditionally used cuff-leak test and may be beneficial to use especially in children with laryngeal edema [39]. When considering the most feared subglottic stenosis formation, the reported incidence varies between 0.25 and 11% of intubated patients in PICU/NICU [67,78,79,80], with no association with cuffed tubes [68,81,82]. The risk factors for subglottic stenosis are in majority also risk factors for PES development: multiple intubation attempts, oversized tracheal tube, duration of mechanical ventilation and low birth weight [8,79,82,83]. In anesthesia and especially emergency airway management, several complications might occur. The most common and serious complication is hypoxia, followed by cardiac dysrhythmias or even cardiac arrest, minor adverse events include dental or mucosal trauma, etc. It has been described that all these adverse events go hand in hand with repeated intubation attempts or the need for reintubation [84], caused by inappropriate tube size or tube misplacement [85]. As the incidence of both, misplacement and high air leak with a need to reintubate is lower in cuffed tubes [3], the use of cuffed tubes might prevent complications associated with emergency airway management.

## 9. Optimal Pediatric Airway Management

Cuff itself is only one part of the optimal airway management. The superior safety aspects of the cuffed tubes were already described. However, several important aspects of airway management partake in the overall airway-related morbidity and mortality. The complex airway management should be stepwise and should include monitoring of vital signs, securing a functional intravenous line, proper head position, functional suction, preoxygenation, and pharmacology-induced sedation (anesthesia induction). In the case of tracheal intubation (infraglottic airway device), where a tube is introduced through *rima glottidis* (between vocal cords), the neuromuscular blocking agents have been used in adult patients to achieve the laryngeal muscle and vocal cords paralysis. Muscle relaxants in pediatric anesthesia are being used less often compared to adults [86,87]. This could be partially explained by the excessive use of supraglottic devices—laryngeal masks. Although tracheal intubation is possible in deep sedation (without muscle relaxants) [88,89,90], muscle relaxation is associated with superior conditions for intubations and better laryngeal view [91]. Despite described benefits, tracheal intubation is often performed without muscle relaxation [92], although a high center and regional variability is described. Only complex optimal and standardized airway management in adult and pediatric patients could further decrease the overall airway-related morbidity and mortality, not the cuffed tubes alone.

## 10. EBM View and Future Perspectives

Cuffed tubes have superior clinical performance with comparable or even lower incidence of complications [7,93]. It can be safely preferred in all pediatric patients after considering the outer and inner diameters of the selected cuffed tube. Further development should decrease the limit for safe use of cuffed tubes also in low birth weight or extremely low birth weight neonates. Despite almost two decades of cuffed tracheal tubes implementation in the anesthesiology and intensive care clinical practice, high-quality EBM data analyzing their long-term use (or the long-term use of cuffed tracheostomy tubes) are still missing. Although strictly recommended to monitor the intracuff pressure and to maintain the cuff pressure below 20 cm H_2_O, the lower intracuff safety pressure limit has not been set. The minimal intracuff pressure could however lead to an increased incidence of regurgitation/aspiration and VAP development. Polyurethane seems to be a material superior to PVC for the cuff; however, tube or cuff coating with silver [94], antibiotics [95] or even nanomaterials to decrease the biofilm formation and VAP incidence could be the source for further tracheal tube research in the future.

## 11. Conclusions

Cuffs have solved many technical problems associated with the uncuffed tubes such as high leak, higher exchange rate, non-accurate capnography or spirometry measurements. Cuffed tubes are being used more and more often in pediatric anesthesiology. The change is much slower in PICUs/NICUs and the emergency department, where tradition is sometimes preventing wider use of cuffed tubes in these settings. According to the EBM data, if properly used, cuff tracheal tubes could/should be the preferred choice for airway management in all pediatric patients and settings, currently with an exception of neonates with an actual body weight below 2700 g. However, in the near future, a cuffed tube size of 2.5 could decrease this limit even further. Currently, no data support the original idea of higher safety of the uncuffed tubes compared to the cuffed ones. The safety issues of the cuffed tubes are directly associated with a proper intracuff pressure management (≤20 cm H_2_O).

## Figures and Tables

**Figure 1 children-09-01490-f001:**
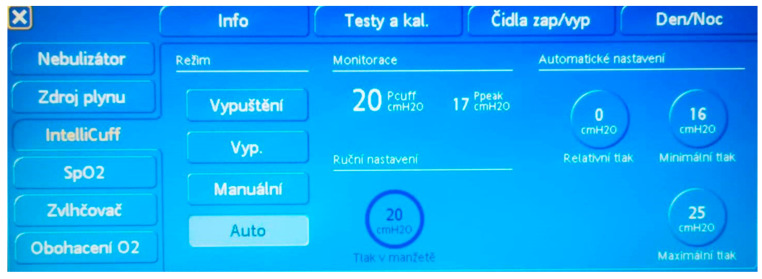
IntelliCuff^®^ setting.

**Figure 2 children-09-01490-f002:**
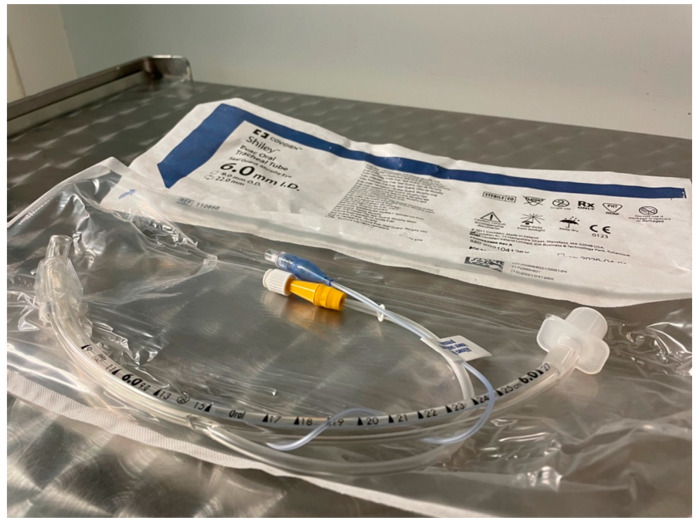
Cuffed tube with the subglottic secretion suctioning.

**Table 1 children-09-01490-t001:** Recommended tube sizes for pediatric patients.

Age	Uncuffed Tracheal Tube	Cuffed Tracheal Tube	Age	Microcuff^®^ Tracheal Tube
	Modified Cole’s Formula	Motoyama Formula		
	ID = (Age/4) + 4	ID = (Age/4) + 3.5		
term-1 year	3.5 mm	3 mm	term to < 8 months	3 mm
<2 years	4 mm	3.5 mm	8 months to < 2 years	3.5 mm
<4 years	4.5 mm	4 mm	<4 years	4 mm
<6 years	5 mm	4.5 mm	<6 years	4.5 mm
<8 years	5.5 mm	5 mm	<8 years	5 mm
<10 years	6 mm	5.5 mm	<10 years	5.5 mm
<12 years	6.5 mm	6 mm	<12 years	6 mm
<14 years	7 mm	6.5 mm	<14 years	6.5 mm
<16 years	7.5 mm	7 mm	<16 years	7 mm

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
