# Peer review of "Airway Management in Pediatric Patients: Cuff-Solved Problem?"

_children, 2022, doi:10.3390/children9101490_

Round 1

Reviewer 1 Report

This is a well written and clinically relevant paper detailing the pros and cons of utilizing cuffed endotracheal tubes in pediatric patients. As this is something I deal with on a daily basis as an anesthesiologist, I think it is good summary, particularly for those who practice outside of the operating room.

A picture depicting the airway anatomy labeling the glottis, subglottis, infraglottis etc would be advantageous for those with less understanding of detailed airway anatomy.

From personal experience, all of my colleagues preferentially use cuffed endotracheal tubes. We do this because of the decreased incidence of tube exchange for resizing, better ability to deliver positive pressure ventilation, ability to reliably monitor end tidal CO2, need for muscle relaxation for certain procedures. We prefer the microcuff tubes because of the benefits as outlined in the paper. However, the chart detailing sizes should be adjusted as it is difficult to read. Below is a more accurate table for sizing microcuff tubes. Please see the attachment.

Unfortunately, we find that our NICU colleagues in particular prefer uncuffed tubes. This provides difficult situations on transfer of care, which I think you should discuss in the section under ICU management. When we get a neonate whose airway can easily accommodate a microcuff tube but present for surgery with an uncuffed tube, we have to then risk changing out the endotracheal tube in a patient who may not be doing well clinically (intestinal perforation, NEC, CDH) and has been intubated for some time. These patients are at high risk during re-intubation, but it must be done in order to properly positive pressure ventilate them for the procedure to be performed.

I appreciate the extensive detail regarding the differences in airway anatomy and the types of tubes. The conclusion could be more robust in discussing how anesthesia has moved to primarily utilizing cuffed endotracheal tubes but other setting particularly NICU and PICU have not. I would remove the last 2 lines of the conclusion because I think there were still be instances where uncuffed tubes are necessary, severe subglottic stenosis but you need adequate length of endotracheal tube. I don’t think one size smaller microcuff tubes will change practice, but rather data to convince ICU physicians that post extubation stridor is not due to cuffed endotracheal tubes is really the answer.

Overall, this is an excellent paper with good clinical significance.

Author Response

We would like to thank the reviewer for their valuable comments. We have tried to implement them in the final version of the article. Also we made small corrections concerning the English language.
The Table 1 was adjusted in size so that it is easier to read. Mention of the need for reintubation and its risk has been added (lines 261-265). 
We slightly rephrased the Conclusion section according to the suggestion made.

Reviewer 2 Report

Excellent review of old habit of uncuffed ET use in under 8 year old pediatric patients.  English editing for sentence composition and general wordiness.  Consider eliminating page 2/5 lines 197 - 216 - paragraph about supraglottic airway devices in the interest of brevity

Author Response

Thank you for your valuable comments. In our article, we wanted to mentioned the supraglottic airways as they should always be considered the device of choice in all situations possible, in order to minimalize the risk of complications as well as the device that can be used in case of failed intubation. Therefore, we decided to implement this paragraph in the article. We made some corrections in the English language.